# Zinc, Zinc Transporters, and Cadmium Cytotoxicity in a Cell Culture Model of Human Urothelium

**DOI:** 10.3390/toxics9050094

**Published:** 2021-04-24

**Authors:** Soisungwan Satarug, Scott H. Garrett, Seema Somji, Mary Ann Sens, Donald A. Sens

**Affiliations:** 1Kidney Disease Research Collaborative, Centre for Health Service Research, University of Queensland Translational Research Institute, Woolloongabba, Brisbane 4102, Australia; 2Department of Pathology, University of North Dakota School of Medicine and Health Sciences, Grand Forks, ND 58202, USA; scott.garrett@med.und.edu (S.H.G.); seema.somji@med.und.edu (S.S.); mary.sens@med.und.edu (M.A.S.); donald.sens@med.und.edu (D.A.S.)

**Keywords:** aza-dC, BSO, cadmium, DNA methylation, glutathione, qRT-PCR, urothelium, ZIP zinc transporters, ZIP14, ZnT zinc transporters, ZnT1

## Abstract

We explored the potential role of zinc (Zn) and zinc transporters in protection against cytotoxicity of cadmium (Cd) in a cell culture model of human urothelium, named UROtsa. We used real-time qRT-PCR to quantify transcript levels of 19 Zn transporters of the Zrt-/Irt-like protein (ZIP) and ZnT gene families that were expressed in UROtsa cells and were altered by Cd exposure. Cd as low as 0.1 µM induced expression of ZnT1, known to mediate efflux of Zn and Cd. Loss of cell viability by 57% was seen 24 h after exposure to 2.5 µM Cd. Exposure to 2.5 µM Cd together with 10–50 µM Zn prevented loss of cell viability by 66%. Pretreatment of the UROtsa cells with an inhibitor of glutathione biosynthesis (buthionine sulfoximine) diminished ZnT1 induction by Cd with a resultant increase in sensitivity to Cd cytotoxicity. Conversely, pretreatment of UROtsa cells with an inhibitor of DNA methylation, 5-aza-2’-deoxycytidine (aza-dC) did not change the extent of ZnT1 induction by Cd. The induced expression of ZnT1 that remained impervious in cells treated with aza-dC coincided with resistance to Cd cytotoxicity. Therefore, expression of ZnT1 efflux transporter and Cd toxicity in UROtsa cells could be modulated, in part, by DNA methylation and glutathione biosynthesis. Induced expression of ZnT1 may be a viable mechanistic approach to mitigating cytotoxicity of Cd.

## 1. Introduction

Exposure to cadmium (Cd) is inevitable for most people because this toxic metal is present in virtually all foodstuffs, cigarette smoke, and polluted air from combustion of fossil fuels and biomass [1,2,3]. Foods that are frequently consumed in large quantities such as rice, potatoes, wheat, leafy salad vegetables, and other cereal crops are the most significant dietary sources [1,2,3]. Cd has no known biological role in humans, but it has a similar ionic radius to that of calcium (Ca) and electronegativity similar to that of zinc (Zn). Consequently, Cd can be taken up by the same transporter systems and pathways the body uses to acquire Ca, Zn, and other essential metals including iron (Fe) and manganese (Mn) [2,4,5,6]. Indeed, members of zinc transporters of the Zrt- and Irt-related protein (ZIP) family, such as ZIP8 and ZIP14, have been shown to mediate Cd uptake by various cell types [7,8,9,10,11,12,13]. High blood Cd has been associated with certain variants of the ZIP8 and ZIP14 genes [14].

In theory, the absorption rate of Cd increases when the body is in short supply of the elements that share absorption and transport pathway mechanisms with Cd, such as Fe, Zn, Mn, and Ca [2,5,6]. It also increases in subjects whose diets are deficient in these elements [2,15]. In mice, a diet deficient in Fe or Ca led to much greater accumulation of Cd in the kidneys than a diet deficient in copper, Zn, or Mn [16]. In humans, high urinary Cd levels, indicative of a high Cd body burden, were associated with low (inadequate) Zn intake in both men and women [15]. In other two studies, low dietary Zn intake was associated with an increased risk of Cd toxicity in kidneys and lungs [17,18]. Low serum Zn levels (<74 μg/dL) were associated with a 3.38-fold increase in risk of estimated glomerular filtration rate (eGFR) <60 mL/min/1.73 m^2^ [17]. High blood Cd levels (>0.53 μg/L) were associated with a 2.04-fold increase in risk of eGFR < 60 mL/min/1.73 m^2^ as compared with blood Cd < 0.18 μg/L. After adjustment for smoking effects, high urinary Cd levels (>0.79 μg/g creatinine) were associated with a 2.48-fold increase in risk of obstructive lung disease as compared with urinary Cd < 0.39 μg/g creatinine [18]. Low Zn intake (<8.35 mg/day) was associated with a 1.89-fold increase in risk of obstructive lung disease as compared with Zn intake above 14.4 mg/day. Thus, dietary Zn intake of 15 mg/day, which was higher than the recommended dietary allowance of 11 mg/day for men and 8 mg/day for women, was required to reduce Cd absorption and limit the body burden of Cd to levels not associated with obstructive lung disease.

In the present study, our aim was to demonstrate the protection against Cd toxicity by simultaneous Zn exposure. In addition, we examined the putative contribution of zinc and zinc transporters in modulating cytotoxicity of Cd. We used a non-tumorigenic human cell line, named UROtsa, since studies in our laboratory have established that the UROtsa cell line is a suitable cell culture model of urinary bladder epithelium [19,20]. Our studies have also indicated that the UROtsa cell line is suitable for studying urothelial cancer; prolonged exposure to 1 µM Cd induced UROtsa cells to transform to cancer cells [21] and the tumors derived from transformed cells displayed gene expression profiles similar to the basal subtype of muscle invasive bladder cancer [22]. Herein, we examined the effects of Cd and simultaneous Zn exposure on expression levels of zinc transporters of ZIP and ZnT gene families expressed by UROsta cells using real-time qRT-PCR. We hypothesized that upregulation of the efflux transporter ZnT1 gene concurrent with downregulation of the influx transporter ZIP genes such as ZIP8 and ZIP14 can prevent Cd accumulation and toxicity. Other research groups have suggested that upregulation of ZnT1, an efflux transporter [23,24] or downregulation of ZIP8, an influx transporter could lead to Cd resistance [25].

## 2. Materials and Methods

### 2.1. Cells and Culture Maintenance

We used the UROtsa cell line derived from the ureter epithelium of a 12-year-old female donor, immortalized with SV40 large T-antigen [18]. It is a non-tumorigenic cell line that displays the phenotypic and morphologic characteristics resembling primary transitional epithelial cells of the bladder [19]. UROtsa cells were cultured in 25 cm^2^ tissue culture flasks in Dulbeco’s modified Eagle’s medium supplemented with 5% vol/vol fetal bovine serum [20,21]. Cells were maintained at 37 °C in humidified incubators with 5% CO_2_/95% atmospheric air. Cells were fed fresh growth medium every three days. At confluence, cells were subcultured at a 1:4 ratio using trypsin-EDTA (0.05%, 0.02%). For experimentation, cells were grown in 6-well plates containing 2 mL growth media per well (35 mm in diameter). Treatment of cultures with Cd [26], Zn, BSO [27], and aza-dC [28] were undertaken when cultures reached 80% confluence. Cd, Zn, BSO, and aza-dC were of highest purity grade (Sigma, St. Louis, MO, USA). For an initial dose-response analysis, five cell cultures were used for each Cd concentration. Expression of ZIP and ZnT genes from the initial experiment agreed with microarray data using the Affymetrix 133 Plus 2.0 [29]. Thus, in later experiments, at least two different batches of cell cultures were used per treatment.

### 2.2. Quantitative Cell Viability Assay

Cell viability assay was based on the capacity of cells to reduce a tetrazolium dye, 3-(4,5-dimethylthiazol-2-yl)-2,5-diphenyltetrazolium bromide (MTT), to formazan [30]. Briefly, forty microliters of 5 mg/mL MTT solution was added to each well of cells grown in 6-well plates, followed by additional 4 h incubation at 37 °C in a 5% CO_2_/95% air atmosphere to allow cellular MTT uptake and MTT reduction [30]. The cells were washed twice with 2 mL of phosphate buffered saline. One ml of acidic propanol (0.1 N HCl in absolute propanol) was added to each well to dissolve formazan that cells generated. The amount of formazan was determined based on absorbance at 570 nm with acidic propanol as the blank.

### 2.3. Quantification of ZIP and ZnT Zinc Transporter Gene Expression

Total RNA was extracted from cells using TRI Reagent^®^ (Molecular Research Center, Inc., Cincinnati, OH, USA). A 2 µL aliquot of RNA sample containing 40 ng of total RNA was subjected to cDNA synthesis using the iScript™cDNA synthesis Kit (Bio-Rad Laboratories, Hercules, CA, USA) in a 20 μL total volume. The cDNA was stored at −20 °C for later analysis. Real-time PCR was performed in triplicate with iQ™SYBR^®^ Green Supermix (Bio-Rad, Hercules, CA, USA) in the iCycler Real-Time Detection System (Bio-Rad Laboratories), using 2 μL of cDNA (equivalent to 4 ng total RNA) in a 20 μL final volume, containing 0.2 μM each of the primers. Primer pairs and PCR conditions for ZIP and ZnT genes assayed were detailed previously [31]. Target amplification was ascertained with post-run melt curve analysis. Quantification was achieved with a standard curve, constructed for each specific gene amplimer. Expression of each zinc transporter gene was normalized to β-actin as the number of transcripts per 1000 or 10,000 β-actin transcripts.

### 2.4. Statistical Analysis

Expression levels of zinc transporter genes and cell viability data were analyzed with SPSS 17.0 (SPSS Inc., Chicago, IL, USA). A Kruskal–Wallis test was used to determine differences in expression levels of an individual zinc transporter gene across four groups of cell cultures, exposed to varied Cd concentrations (five cultures for every Cd concentration and control). The Spearman’s rank correlation test was used to evaluate a correlation between expression levels of an individual zinc transporter gene and Cd exposure concentrations. Differences in means for pairs of variables such as untreated control versus treated cells were determined by a Mann–Whitney test. *P*-values ≤ 0.5 were considered to identify statistically significant differences.

## 3. Results

### 3.1. Dose-Response Analysis of Effects of Cadmium on Expression Levels of Zinc Transporters

Expression levels of ZIP and ZnT zinc transporters by UROtsa cells 24 h after exposure to 0, 1, 2, and 4 µM Cd concentrations are shown in Table 1. The expression level as number of transcripts of each individual zinc transporter gene was relative to 1000 or 10,000 β-actin gene transcripts.

At basal state (0 µM Cd), ZnT7 was most abundantly expressed among the ZnT family members, followed by ZnT5 and ZnT1 with on average of 758, 510, and 365 transcript copies per 1000 transcripts of β-actin, respectively. On-average transcript copies of ZnT4, ZnT6, and ZnT10 were 65, 51, and 11 and 1 per 1000 transcripts of β-actin while ZnT2 and ZnT3 were expressed between 6 and 15 transcripts per 100,000 β actin transcripts. For ZIP family, ZIP7 was most abundantly expressed at basal state followed by ZIP14, ZIP6, ZIP1, ZIP10, and ZIP3A with on average of 204, 146, 92, 82, 54, and 19 transcripts per 1000 β-actin transcripts, respectively. ZIP2, ZIP3B, and ZIP8 were expressed between 1 and 4 transcripts per 1000 β-actin transcripts. ZIP 4 and ZIP5 were expressed between 1 and 6 transcripts per 10,000 β-actin transcripts.

Exposure to varying Cd concentrations for 24 h affected expression levels of multiple zinc transporters. Shown in Table 2 are results of the Spearman’s rank correlation analysis to globally assess Cd effects on expression levels of 19 zinc transporters (8 ZnT genes and 11 ZIP genes) measured.

For most zinc transporters, their transcript levels showed an inverse relationship with concentrations of Cd to which cells were exposed. For the ZnT family, transcript levels for ZnT3 were decreased with increasing Cd concentrations (*r* = −0.28, *p* = 0.03) as were ZnT4 transcript levels (*r* = −0.52, *p* < 0.001). Likewise, ZIP1, ZIP3A, ZIP5, and ZIP7 gene expression levels showed an inverse correlation with Cd exposure levels with *r* and *p*-values of −0.29 (*p* = 0.02), −0.33 (*p* = 0.01), −0.27 (*p* = 0.05), and −0.42 (*p* = 0.001), respectively. A particularly strong inverse correlation was seen between Cd exposure levels and transcript levels of two ZIP zinc transporters, ZIP2 (*r* = −0.69, *p* < 0.001) and ZIP10 (*r* = −0.68, *p* < 0.001). Distinct from all other zinc transporters, ZnT1 gene transcript levels showed a positive correlation with exposure Cd concentrations (*r* = 0.51, *p* < 0.001).

### 3.2. Time-Course and Dose-Response Analyses for Induced Expression of the ZnT1 Gene by Cadmium

Cd at 1 µM concentration caused ZnT1 transcript levels to increase from 8 h, with a further increase occurring 16 h later, reaching a plateau at 36 h (Figure 1A). In contrast, Cd at 2 and 4 µM concentrations did not induce expression of ZnT1 gene. The ZnT1 gene expression levels were increased progressively with an increment of Cd^2+^ concentrations from 0.1 to 1.5 µM (Figure 2B). The highest fold induction within 24 h was 8.8, achieved with 1.5 µM Cd concentration. Exposure to 0.25 µM Cd for 48 h induced ZnT1 expression to the same extent achieved with 24-h exposure to 1.5 µM Cd. Of note, 24-h exposure to 1 µM Cd resulted in a 7.5-fold increase in ZnT1 expression. However, ZnT1 fold induction declined from 7.5 at 24 h to 1.4 at 48 h, suggesting toxicity of 1 µM Cd when Cd exposure was continued for a further 24 h.

Exposure to 0.05 and 0.5 µM concentrations of Cd for 24 h did not reduce cell viability, while exposure to 1, 1.5, 2.5, 4, and 5 µM of Cd for 24 h decreased cell viability by 15%, 29%, 57%, 66%, and 74%, respectively (Figure 2A). Thus, 2.5 µM Cd was considered to be a 50% lethal dose (LD_50_) of Cd in UROtsa cells. Exposure to Cd at the LD_50_ level (2.5 µM) for 48 h together with Zn at 10, 25, and 50 µM concentrations prevented loss of cell viability caused by Cd. This was evident from the viability of 80% among those with concurrent Zn exposure as compared with 14% of those exposed to Cd alone with equivalent Cd molar concentrations (Figure 2B).

Of note, an average cell viability of 80% among cells exposed to both 2.5 µM Cd^2+^ and 10, 25, or 50 µM Zn was the same as those exposed only to 1 µM Cd for 24 h (Figure 2B). Intriguingly, exposure to 2.5 µM Cd concurrent with 10 or 25 µM Zn increased ZnT1 expression by the same extent as exposure to 1 µM Cd alone (Figure 2C). Expression levels of ZnT1, ZnT5, ZnT7, ZIP8, and ZIP14 genes were not affected by exposure to Zn at 10 or 25 µM Zn. Simultaneous exposure to 10 µM Zn and 2.5 µM Cd increased expression levels of ZnT5, ZIP8,and ZIP14 genes by the same extent as exposure to 1 µM Cd alone (Figure 2D).

### 3.3. Effects of Pretreatment with BSO and Aza-dC on Cytotoxicity of Cadmium and Expression of ZnT1 Gene

As expected, inhibition of glutathione biosynthesis by BSO treatment 24 h prior to Cd exposure increased sensitivity to Cd (Figure 3A). Such sensitivity increase was seen at all three BSO concentrations tested (100, 500, and 1000 µM). At 0.5 µM Cd, the average cell viability of the BSO-treated cells was 72%, compared with 94% of untreated cells. An average cell viability of the three BSO-treated cultures in cells exposed to the LD_50_ level (2.5 µM Cd^2+^) was 19% as compared with 54% in untreated cells. BSO pretreatment affected ZnT1 by Cd. In untreated cells, 1 µM Cd increased ZnT1 expression by 7.5, and it fell to 4.2, 3.7, and 2.7 in cells treated with 100, 500, and 1000 µM BSO, respectively (Figure 3B).

Inhibition of DNA methylation by 5-aza-2’-deoxycytidine (aza-dC) treatment 24 h prior to Cd exposure produced results opposite to BSO treatment. Tolerance or resistance to Cd toxicity was seen among cells treated with aza-dC at all three aza-dC concentrations tested (2.5, 5, and 10 µM) (Figure 3C). At 2.5 µM Cd, an average cell viability of aza-dC treated cells was 75%, compared with 54% of untreated cells (*p* < 0.5). At 5 µM Cd^2+^, the viability of untreated cells was 27% only, compared with 80%, 56.6%, and 57.1% in cells pretreated with 2.5, 5, or 10 µM aza-dC, respectively (*p* < 0.05). For aza-dC-treated UROtsa cells, LD_50_ for Cd was increased by two-fold (from 2.5 to 5 µM Cd). Aza-dC treatment did not affect expression of ZnT1 nor the extent of ZnT1 induction by 2.5 or 5 µM Cd (Figure 3D).

### 3.4. Other Notable Effects of BSO or Aza-dC Pretreatment

Figure 4 depicts the influence of BSO or aza-dC pretreatment on the changes in expression levels of some zinc transporters caused by subsequent Cd exposure.

Cd at 1 µM did not affect expression of ZnT5, ZnT7, ZIP8, or ZIP14 in untreated control cells (Figure 4A). However, in cells treated with 1000 µM BSO, Cd at 1 µM caused a reduction in expression levels of these five zinc transporters. Cd at 1 µM also caused a reduction in expression levels of ZnT5 and ZnT7 in cells treated with 500 µM BSO, but it did not affect ZIP8 or ZIP14 transcript levels. In contrast, expression levels of ZnT5, ZnT7, ZIP8, and ZIP14 in aza-dC-treated cells at all three concentrations (250, 500, and 1000 µM) were not altered by 2.5 µM Cd (Figure 4B). A higher Cd (5 µM) increased expression levels of ZIP14 in cells treated with 2.5 µM and 10 µM aza-dC (Figure 4C).

Figure 4D depicts the evolution of Cd resistance phenotype induced by aza-dC treatment. In this experiment, UROtsa cell cultures were treated with 2.5 µM aza-dC and they were exposed to varying Cd concentrations 24 h later. Then, they were incubated for a further 24, 48, or 72 h period. Treatment of cells with 2.5 µM aza-dC alone did not affect cell viability, assessed at 24, 48, and 72 h, but Cd at 1, 2, 2.5, and 5 µM caused a 25–40% reduction in cell viability within 48 h. Intriguingly, at 72 h, the viability of all Cd exposed cell cultures were not different from that of cell cultures treated with 2.5 µM aza-dC alone. Thus, resistance to Cd toxicity appeared to be acquired fully in the last 24 h.

## 4. Discussion

Expression levels of individual zinc transporter genes, in the basal state, varied widely between the two gene families of zinc transporters, as did expression levels among members in each family (Table 1). ZIP8, ZIP14, and ZnT1 zinc transporter genes were expressed at 2.1, 146, and 365 copies per 1000 copies of β-actin gene, respectively. Lower expression levels of ZIP8 than ZIP14 suggested that ZIP14 and ZnT1 could potentially be involved in Cd accumulation and its cytotoxicity in this human urothelial cell line. ZIP8 protein expressed by UROtsa cells was detectable by Western blot analysis [32]. We postulated that a reduction in expression of zinc transporters that are responsible for Cd uptake, ZIP8 and ZIP14 included, in conjunction with an increase in expression of Cd excluding transporter such as ZnT1 would produce UROtsa cells that would be resistant to Cd toxicity.

It has been shown in a previous study using a rat liver epithelial cell line (TRL 1215) that pretreatment of cells with cyproterone, a synthetic steroidal antiandrogen with a structure related to progesterone, decreased sensitivity to Cd through a decreased accumulation of Cd [33]. However, the molecular basis for a decrease in Cd accumulation was not investigated. It was shown in another study that silencing of ZnT1 expression increased Cd accumulation and enhanced Cd toxicity [23]. A decrease in Cd accumulation together with a decrease in ZIP8 expression, assessed by ZIP8 mRNA and ZIP8 protein levels was seen in metallothionein-null cells that were resistant to Cd toxicity [34], and epigenetic silencing of the ZIP8 gene by DNA hypermethylation was found to be involved in the downregulation of ZIP8 expression observed [34]. It was suggested that downregulation of ZIP8 was a common feature of cells resistant to Cd [35]. Thus, DNA hypermethylation appeared to be the molecular mechanism by which the ZIP8 gene is downregulated.

In a correlation analysis (Table 2), an inverse relationship was seen between Cd exposure concentrations and expression levels of most genes examined. The correlation was especially strong for ZIP2 (*r* = −0.69, *p* < 0.001) and ZIP10 genes (*r* = −0.68, *p* < 0.001). A positive correlation was seen only between ZnT1 gene expression and Cd exposure concentrations (*r* = 0.51, *p* < 0.001). We considered it as initial evidence for a selective induction ZnT1 by Cd, which may represent a defense mechanism of UROtsa cell use to prevent Cd and Zn accumulation to toxic levels. Time-course and dose-response analyses have confirmed that Cd concentrations between 0.1 and 1.5 µM increased expression of the ZnT1 gene. Of note, at Cd concentrations between 0.05 and 0.5 µM concentration, cell viability was not affected. The highest fold ZnT1 induction of 8.8 occurred within 24 h of exposure to 1.5 µM Cd (Figure 1A,B). Among 19 zinc transporters examined, the zinc efflux transporter ZnT1 was found to be induced by Cd in both time- and dose-dependent manners. The magnitude of ZnT1 induction correlated with sensitivity/resistance to Cd cytotoxicity.

Evidence for the protective role of ZnT1 upregulation has emerged from a series of experiments involving simultaneous Zn exposure, BSO, and aza-dC pretreatments. Simultaneous exposure to Zn at 4-, 10- and 40-times higher molar concentrations than that of Cd prevented loss of cell viability, indicative of Cd cytotoxicity (Figure 2A,B). Under these experimental conditions, expression of the ZnT1 gene was increased by eight-fold (Figure 2C). Exposure to 10 µM Zn together with 2.5 µM Cd increased expression levels of ZnT5, ZIP8, and ZIP14 genes by the same extent as did an exposure to 1 µM Cd alone (Figure 2D). These data could be interpreted to suggest that concurrent Zn exposure at 4-times higher molar concentration than Cd resulted in an “effective” intracellular Cd concentration similar to that of exposure to 1 µM Cd. The protective effect of high-dose Zn exposure concurrent with Cd exposure was demonstrated in a study using human retinal pigment epithelial cell lines [36], but the mechanisms of such protection such as expression of zinc transporters were not measured in the study nor did all other studies examining protective effects of concurrent Zn treatment in cell cultures.

BSO treatment caused a reduction in ZnT1 inductive responses coincidental with an increase in sensitivity to Cd (Figure 3B). Cd at 2.5 µM caused an 81% reduction in cell viability in BSO-treated cells, compared with 54% in untreated cells (Figure 3A). The viability loss in BSO-treated cells occurred in conjunction with a fall in ZnT1 induction from 7.5-fold to 4.2-, 3.7- and 2.7-fold in cells treated with 100, 500, and 1000 µM BSO, respectively (Figure 3B). Increased sensitivity to Cd following treatment with BSO could partially be attributed to diminished Cd efflux, a consequence of depressed ZnT1 induction.

BSO treatment is known to perturb both GSH biosynthesis and DNA methylation [27]. Thus, induction of ZnT1 could be a consequence of depletion of GSH or alteration of DNA methylation or both. Previously, ZIP8 expression was shown to be modulated by GSH concentrations [25]. In the present study, we observed reduced expression of ZIP8 together with reduced expression of ZnT5, ZnT7, and ZIP14 in cells treated with BSO for 24 prior to exposure to 1 µM Cd (Figure 4A). To distinguish these possibilities, we compared results from BSO treatment with those from aza-dC treatment experiments.

Unlike BSO treatment, pretreatment of UROtsa cells with aza-dC for 24 h prior to Cd exposure resulted in resistance to Cd cytotoxicity (Figure 3C). Aza-dC treatment caused a two-fold increase in Cd resistance while it had no effects on ZnT1 inductive responses or expression levels of ZnT5, ZnT7, and ZIP8. Thus, Cd-induced expression of ZnT1 appeared to be influenced by GSH concentrations which reflect cell redox state. Likewise, effects of Cd on expression levels of ZnT5, ZnT7, and ZIP8 genes may vary with cellular redox status but not with gene methylation status. However, the effects of Cd on ZIP14 gene expression seemed to depend on both GSH and the methylation state of the ZIP14 gene. Further research is warranted to investigate epigenetic regulation of the expression of zinc transporters that may modulate Cd toxicity.

## 5. Conclusions

Expression of Cd cytotoxicity in UROtsa cells and the expression level of ZnT1 zinc transporter, known to medicate efflux of both Zn and Cd, could be modulated, at least in part, by DNA methylation state and glutathione biosynthesis. A diminished ZnT1 induction may account for an increased sensitivity to Cd cytotoxicity in cells treated with BSO prior to Cd exposure. Conversely, ZnT1 induction that remains impervious in aza-dC-treated cells may account for resistance to Cd cytotoxicity.

## Figures and Tables

**Figure 1 toxics-09-00094-f001:**
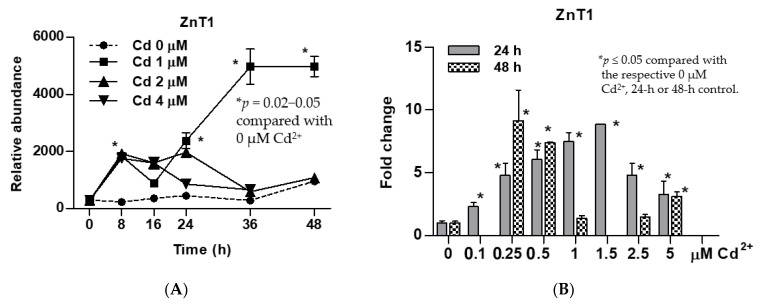
Time-course and dose-response analyses for an induced expression of the ZnT1 zinc transporter gene by cadmium. (**A**) Expression levels of the ZnT1 gene as a function of exposure durations and Cd^2+^ concentrations; (**B**) expression levels of the ZnT1 gene 24 h and 48 h after exposure to varying Cd^2+^ concentrations. Fold change or a ratio was defined as the number of transcripts of a given gene relative to β-actin in control cells divided by number of transcripts of the same gene in treated cells relative to β-actin. ZnT1 expression in (**A**,**B**) were from different batches of cells. An effect of 48-h exposure was not assessed for 0.1 and 1.5 µM Cd concentrations.

**Figure 2 toxics-09-00094-f002:**
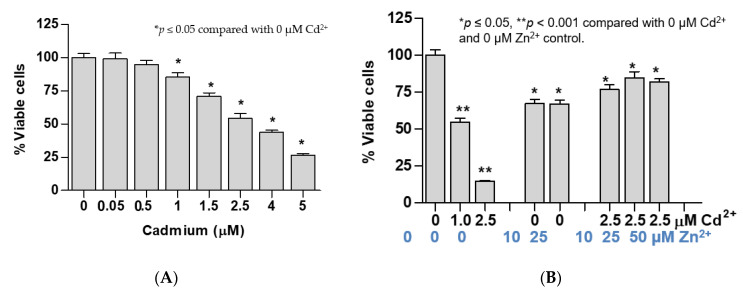
Effects of exposure to cadmium alone or cadmium plus zinc on cell viability and expression levels of ZIP and ZnT zinc transporter genes. (**A**) Change in cell viability following exposure to only cadmium; (**B**) zinc protection against cadmium-induced loss of cell viability; (**C**) expression levels of ZnT1 gene in cells exposed to cadmium and zinc; (**D**) expression levels of ZnT5, ZnT7, ZIP8, and ZIP14 genes in cells exposed to cadmium and zinc. Fold change or a ratio was defined as number of transcripts of a given gene relative to β-actin in control cells divided by number of transcripts of the same gene in treated cells relative to β-actin. Cell viability data in (**A**,**B**) were assessed with different batches of cells. Expression of ZnT1 in (**C**) and expression of other ZnT and ZIP transporters in (**D**) were assessed with an identical cell batch.

**Figure 3 toxics-09-00094-f003:**
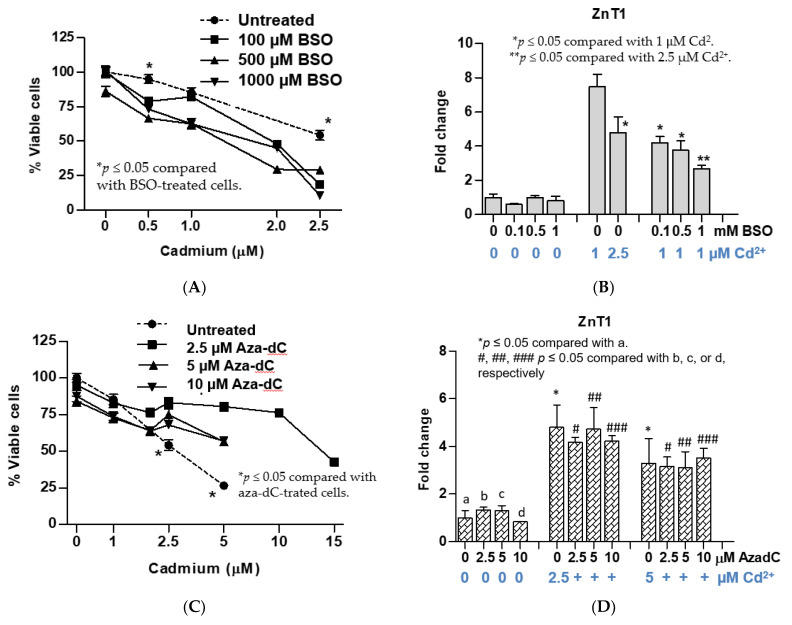
Effects of cadmium in UROtsa cells pretreated with BSO or aza-dC. (**A**) An analysis of cell viability reduction; (**B**) changes in ZnT1 expression in BSO-treated UROtsa cells; (**C**) an analysis of resistance to loss of cell viability; (**D**) changes in ZnT1 expression in aza-dC-treated UROtsa cells. Fold change or a ratio was defined as number of transcripts of a given gene relative to β-actin in control cells divided by number of transcripts of the same gene in treated cells relative to β-actin. Cell viability in (**A**) and ZnT1 expression in (**B**) were assessed with an identical cell batch. Cell viability in (**C**) and ZnT1 expression in (**D**) were assessed with an identical cell batch.

**Figure 4 toxics-09-00094-f004:**
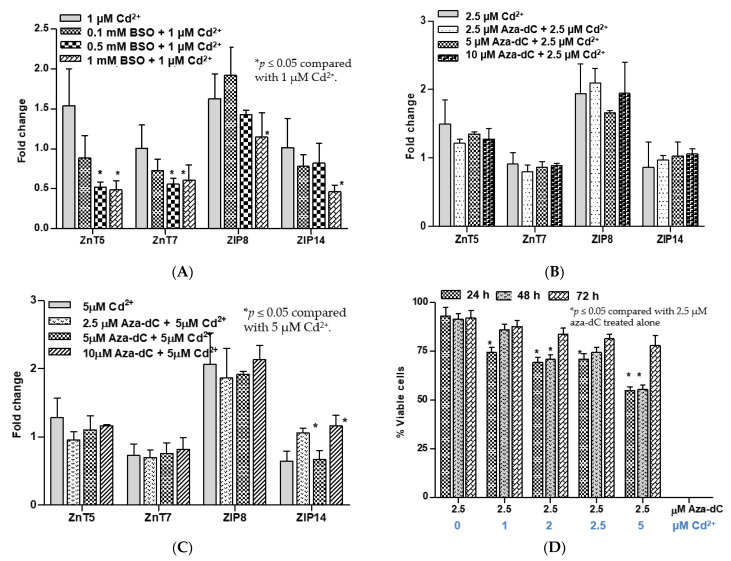
Effects of cadmium on expression of selected zinc transporters and viability of BSO- and aza-dC-treated cells. (**A**) Changes in expression levels of ZnT5, ZnT7, ZIP8, and ZIP14 in BSO-treated cells caused by 1 µM Cd^2+^; (**B**) changes in expression levels of ZnT5, ZnT7, ZIP8, and ZIP14 in aza-dC-treated cells caused by 2.5 µM Cd^2+^; (**C**) changes in expression levels of ZnT5, ZnT7, ZIP8, and ZIP14 in aza-dC-treated cells caused by 5 µM Cd^2+^. Effects of 24, 48, and 72 h of cadmium exposure on viability of UROTsa cells pretreated with 2.5 µM aza-dC (**D**). Fold change or a ratio was defined as number of transcripts of a given gene relative to β-actin in control cells divided by number of transcripts of the same gene in treated cells relative to β-actin. Expression profiles in (**A**) were based on RNA samples from cells in Figure 3B. Expression profiles in (**B**,**C**) were based on RNA samples from cells in Figure 3D. An effect of aza-dC treatment duration in (**D**) was assessed with a different batch of cells.

**Table 1 toxics-09-00094-t001:** Dose-response analysis of effects of cadmium on expression levels of zinc transporter genes in cell culture model of human urothelium (UROtsa) cells.

Zinc Transporters	Cadmium Concentration, µM				*p*-Values
	0	1	2	4	
SLC30A family					
ZnT1	365 ± 38	3007 ± 465	1434 ± 146	1216 ± 153	<0.001
ZnT2	0.06 ± 0.01	73 ± 15	16 ± 1.9	11 ± 1.5	<0.001
ZnT3	0.15 ± 0.01	0.24 ± 0.05	0.10 ± 0.02	0.10 ± 0.02	0.03
ZnT4	11.4 ± 0.8	10 ± 1	8.7 ± 1	6.2 ± 0.5	0.001
ZnT5	510 ± 30	1038 ± 132	495 ± 54	568 ± 91	0.001
ZnT6	65 ± 8	77± 6	63 ± 13	57 ± 12	0.174
ZnT7	758 ± 76	1007 ±136	706 ± 44	488 ± 63	0.02
ZnT10	1.1 ± 0.2	2.4 ± 0.2	1.7 ± 0.2	1.1 ± 0.1	<0.001
SLC39A family					
ZIP1	82 ± 9	99 ± 15	55 ± 10	59 ± 12	0.02
ZIP2	1.2 ± 0.1	0.8 ± 0.2	0.4 ± 0.1	0.2 ± 0.03	<0.001
ZIP3A	19 ± 1	23 ± 2.3	17 ± 1.7	14 ± 1.3	0.01
ZIP3B	4.1 ± 0.2	6.2 ± 0.7	4.4 ± 0.2	4.2 ± 0.4	0.02
ZIP4	0.06 ± 0.01	0.06 ± 0.01	0.04 ± 0.01	0.06 ± 0.01	0.40
ZIP5	0.01 ± 0.001	0.01 ± 0.003	0.01 ± 0.002	0.01 ±.002	0.28
ZIP6	92 ± 8	133 ± 12	80 ± 9	75 ± 10	0.001
ZIP7	204 ± 25	342 ± 69	149 ± 32	94 ± 21	<0.001
ZIP8	2.1 ± 0.2	2.6 ± 0.3	2.0 ± 0.2	2.7 ± 0.4	0.50
ZIP10	54 ± 4	30 ± 8	14 ± 3	14 ± 3	<0.001
ZIP14	146 ± 19	218 ± 24	158 ± 26	128 ± 19	0.01

Numbers are values for mean ± standard error of mean for the number of transcripts for each individual zinc transporter gene relative to 1000 transcripts of the β-actin gene. *p*-values were based on a Kruskal–Wallis test, and *p*-values ≤0.05 were considered to identify statistically significant differences in expression levels of an individual zinc transporter gene across four groups, exposed to varying Cd^2+^ concentrations.

**Table 2 toxics-09-00094-t002:** Correlations between expression levels of various zinc transporter genes and exposure concentrations of cadmium.

SLC30A Family	Concentration of Cd^2+^ (0, 1, 2, 4 µM)		SLC39A Family	Concentration of Cd^2+^ (0, 1, 2, 4 µM)	
	*r*	*p*-Values		*r*	*p*-Values
ZnT1	0.51	<0.001	ZIP1	−0.29	0.02
ZnT2	0.10	0.33	ZIP2	−0.69	<0.001
ZnT3	−0.28	0.03	ZIP3A	−0.33	0.01
ZnT4	−0.52	<0.001	ZIP3B	0.10	0.44
ZnT5	−0.01	0.92	ZIP4	−0.08	0.52
ZnT6	−0.07	0.56	ZIP5	−0.27	0.05
ZnT7	−0.24	0.06	ZIP6	−0.19	0.13
ZnT10	0.11	0.40	ZIP7	−0.42	0.001
			ZIP8	0.04	0.622
			ZIP10	−0.68	<0.001
			ZIP14	−0.03	0.84

*R*, Spearman rank correlation coefficient *rho*, which indicates the strength of a correlation between expression levels of an individual zinc transporter and exposure concentrations of Cd. The *p*-values of ≤ 0.5 were considered to identify statistically significant relationships between these two parameters.

## Data Availability

Data are contained within the article.

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
