# Peer review of "Zinc, Zinc Transporters, and Cadmium Cytotoxicity in a Cell Culture Model of Human Urothelium"

_toxics, 2021, doi:10.3390/toxics9050094_

Round 1

Reviewer 1 Report

This is a very interesting article about the protection role of zinc as well as zinc transporters
against cadmium in in vitro model using human ureter epithelium cells.
The manuscript is very well written and prepared. The results are interesting, well documented
and the discussion of the above results is based on new references.

In my opinion this MS is suitable for publication after minor revisions:

The Authors should delete the sentences from lines 70 to 75 from the last part of the Introduction. 

In Table 1 the statistical differences should be marked (different letters, an asterisk, etc). 

In my opinion, the description of the figures in the text should be deleted (lines 170-172, 188-190, 217-218, 246-248). 

In Figure 3D some statistics should be marked.  

In Figure 1 the data should be checked between A and B. 

Author Response

This is a very interesting article about the protection role of zinc as well as zinc transporters against cadmium in in vitro model using human ureter epithelium cells.
The manuscript is very well written and prepared. The results are interesting, well documented and the discussion of the above results is based on new references.

In my opinion this MS is suitable for publication after minor revisions:

Response to top comments: We thank the Reviewer for evaluation of the scientific merit of our paper and for advice, comments and suggestion to further improve our paper.

Point 1: The Authors should delete the sentences from lines 70 to 75 from the last part of the Introduction. 

Response: As advised, we have deleted the referred statements.

Point 2: In Table 1 the statistical differences should be marked (different letters, an asterisk, etc). 

Response: In the original manuscript, the p-values indicating statistical significance in expression levels of influx/efflux zinc transporters are given in the last column of Table 1. The p-values were based derived by Kruskal-Wallis test which is suitable for an evaluation of an overall effect of Cd-dose levels used. We retain this presentation format in a revised version of our paper.  

Point 3: In my opinion, the description of the figures in the text should be deleted (lines 170-172, 188-190, 217-218, 246-248). 

Response: The description of figures to which the Reviewer referred to have all deleted. 

Point 4: In Figure 3D some statistics should be marked.  

Response: Statistical test results have now been provided for Figure 3D.

Point 5: In Figure 1 the data should be checked between A and B. 

Response: Results in Figures 1A and 1B are from two separate experiments.  In Figure 1A, ZnT1 expression levels were of 24-h exposure to 0, 1, 2, 4 µM Cd, where the only 1 µM Cd increased ZnT1 expression. We, then conducted new series of experiments covering lower ranges of Cd concentrations (0, 0.1, 0.25, 0.5, 1 µM Cd) which are different from those shown in Figure 1A. Data in Figure 1 B indicated Cd as low as 0.1 µM induced ZnT1 expression.  We have corrected an error made in abstract on the lowest Cd concentration that increased ZnT1. An effect of 48-h exposure duration was done for certain Cd concentration only as shown Figure 1B. 

Reviewer 2 Report

Major comments:

  • The authors described the abstract of the study in the last paragraph in Introduction. However, the significance and originality of their focus and hypothesis are unclear. Why and how could this study contribute to the issues raised in the previous paragraphs in Introduction? It should be explained in a reasonable and convincing manner in Introduction.
  • Table 1 and 2 should be presented in graph format.
  • In Figure 1, the expression levels of ZnT1 do not seem to be consistent in the same conditions in (A) and (B), for example, 1uM Cd exposure for 24h and 48h.
  • There are missing data in Figure 1 (B), for example 1.5uM Cd for 48h, and the positions of Asterisks are unclear and misleading.
  • In Figure 2 (A) and (B), viabilities of cells are not consistent, e.g., 2.5uM Cd2.
  • The authors claimed that they found that Cd as low as 0.25 µM induced expression of ZnT1. It may be based on Figure 1 (B), but there is also an asterisk on the bar at 0.1 uM Cd exposure.
  • In Figure 3 (A) and (B), why are there asterisks on plots of untreated conditions?
  • In Figure 3 (D), it does not seem to be right not to see any significance on these data.
  • An DNA methylation inhibitor had some effects on Cd cytotoxicity, but it is not enough to state that DNA methylation state are critical to the Cd cytotoxicity. It is just a speculation, because there are no direct proofs presented and inhibitors can have many side effects.
  • The authors claimed that the diminished ZnT1 induction may account for an increased sensitivity to Cd cytotoxicity in cells treated with BSO prior to Cd exposure, but ZnT1 seemed to be induced by BSO as show in in Figure 3 (B).

Author Response

We thank the Reviewer for her/his thorough evaluation and for comments and suggestions.  We have provided below point-by-point response to the issues raised.

Major comments:

Point 1: The authors described the abstract of the study in the last paragraph in Introduction. However, the significance and originality of their focus and hypothesis are unclear. Why and how could this study contribute to the issues raised in the previous paragraphs in Introduction? It should be explained in a reasonable and convincing manner in Introduction.

Response: The referred statement has been deleted, and we have rewritten part of the Introduction to indicate the purposes and working hypothesis (lines 60-71). To the best of our knowledge the present study has shown for the first time that expression of the zinc efflux transporter, ZnT1 is induced by Cd. It also first report of possible epigenetic regulation of zinc transporters other than ZIP8 that has been shown by other research group.

Point 2: Table 1 and 2 should be presented in graph format.

Response: We believe that table format is suitable for quantitative data (exact copy number of transcripts for 19 zinc transporters of two gene families). They form database for comparing expression of zinc transporter in other cell types. We have chosen to present results of a correlative analysis in table format because this format gives information on the strength of a correlation between transcript levels of 19 zinc transporters and Cd exposure concentrations together with statistical significance levels of those correlations in about half a page.

Point 3: In Figure 1, the expression levels of ZnT1 do not seem to be consistent in the same conditions in (A) and (B), for example, 1uM Cd exposure for 24h and 48h.

Response: Results in Figures 1A and 1B are from two separate experiments.  Expression levels of ZnT1 in Figure 1A were of 24-h exposure to 0, 1, 2, 4 µM Cd where the results indicate that only 1 µM Cd induced ZnT1. We, then conducted new series of experiments covering lower ranges of Cd concentrations (0, 0.1, 0.25, 0.5, 1 µM Cd) different from those shown in Figure 1A. Data in Figure 1 B indicated Cd as low as 0.1 µM induced ZnT1 expression.  We have corrected an error made in abstract on the lowest Cd concentration that increased ZnT1 (line 14).

Point 4: There are missing data in Figure 1 (B), for example 1.5uM Cd for 48h, and the positions of Asterisks are unclear and misleading.

Response: We assessed an effect of 48-h exposure duration for certain Cd concentration only as shown Figure 1B. 

Point 5: In Figure 2 (A) and (B), viabilities of cells are not consistent, e.g., 2.5uM Cd2.

Response: Results in Figures 2A and 2B were from separate experiments conducted with different batches of cells.  Variability in results is not unexpected in most work involving cells in culture. Despite non-identical results from two independent experiments, our data show a protective effect of simultaneous Zn exposure on cell viability.  

Point 6: The authors claimed that they found that Cd as low as 0.25 µM induced expression of ZnT1. It may be based on Figure 1 (B), but there is also an asterisk on the bar at 0.1 uM Cd exposure.

Response: Thank you for pointing out an error in our data presentation.  As we have indicated in response to point 5 above that we have now rectified an error made in abstract on the lowest Cd concentration that increased ZnT1 to be 0.1 µM not 0.25 µM.

Point 7: In Figure 3 (A) and (B), why are there asterisks on plots of untreated conditions?

Response: In Figure 3A, we have corrected the text box explaining “*” to read, * p ≤ 0.05 compared with BSO-treated cells. In Figure 3C (not 3B), we have corrected the text box explaining “*” to read, * p ≤ 0.05 compared with aza-dC-treated cells.

Point 8: In Figure 3 (D), it does not seem to be right not to see any significance on these data.

Response: Statistical test results have now been provided for the referred Figure.

Point 9: An DNA methylation inhibitor had some effects on Cd cytotoxicity, but it is not enough to state that DNA methylation state are critical to the Cd cytotoxicity. It is just a speculation, because there are no direct proofs presented and inhibitors can have many side effects.

Response: In revised version we have rewritten the pertaining statements (lines 21-23, 338-340).

Point 10: The authors claimed that the diminished ZnT1 induction may account for an increased sensitivity to Cd cytotoxicity in cells treated with BSO prior to Cd exposure, but ZnT1 seemed to be induced by BSO as show in Figure 3 (B).

Response: We thank the Reviewer for pointing out that we missed labelling results in Figure 3B. We have now rectified the errors in the text box explaining statistical test results. In the original manuscript, we noted a possible effect of the highest BSO dose at 1000 µM on expression of ZnT5, ZnT7, ZIP8 and ZIP14 exposed to 1 µM Cd 24h after BSO treatment shown in Figure 4A (lines 245-246).

Reviewer 3 Report

The manuscript entitled, “Zinc, Zinc Transporters and Cadmium Cytotoxicity in a Cell 2 Culture Model of the Human Urothelium” examine the putative contribution of zinc and zinc transporters in modulating cadmium toxicity in UROtsa cell line. The authors have done a thorough expression profiling of 19 known zinc transporters belonging to two different families (SLC30A and SLC39A) at the transcript level at increasing doses of cadmium exposure. In addition, they provide experimental evidence to support the idea that zinc treatment can reverse the cytotoxic effects of cadmium exposure as well as the changes in expression profile of several zinc transporters. Furthermore, they demonstrate that that cellular DNA methylation state and redox state are important modulators of both cytotoxic effects of cadmium exposure, as well as zinc transporter expression profile. The manuscript is clearly written, and the conclusions follow from the data. The figures are clear and easy to follow.

The manuscript can improve substantially by the addition of a few new experiments. The suggestions for these experiments are appended herewith:

  1. The authors have provided data at the transcript level. However, changes in transcript level does not automatically lead to changes at the protein level. The authors should demonstrate (at least for ZnT1) that cadmium exposure leads to a change in the steady state protein expression parallel to that of mRNA. The authors should also provide immunoblot data for ZnT1 for the zinc, BSO and Aza-dC treatment experiments.
  2. It appears from the manuscript that the authors are suggesting that cadmium toxicity (by itself, or in presence of zinc/BSO/Aza-dC) is modulated by changes in zinc transporter expression and consequently on the concentration of intracellular cadmium. The authors need to measure the level of intracellular cadmium under each of these conditions to empirically validate this hypothesis.

Author Response

The manuscript entitled, “Zinc, Zinc Transporters and Cadmium Cytotoxicity in a Cell Culture Model of the Human Urothelium” examine the putative contribution of zinc and zinc transporters in modulating cadmium toxicity in UROtsa cell line. The authors have done a thorough expression profiling of 19 known zinc transporters belonging to two different families (SLC30A and SLC39A) at the transcript level at increasing doses of cadmium exposure. In addition, they provide experimental evidence to support the idea that zinc treatment can reverse the cytotoxic effects of cadmium exposure as well as the changes in expression profile of several zinc transporters. Furthermore, they demonstrate that that cellular DNA methylation state and redox state are important modulators of both cytotoxic effects of cadmium exposure, as well as zinc transporter expression profile. The manuscript is clearly written, and the conclusions follow from the data. The figures are clear and easy to follow.

The manuscript can improve substantially by the addition of a few new experiments. The suggestions for these experiments are appended herewith:

Response to top comments: We thank the Reviewer for his/her insightful comments. We have provided below point-by-point response to the issues raised.

Point 1: The authors have provided data at the transcript level. However, changes in transcript level does not automatically lead to changes at the protein level. The authors should demonstrate (at least for ZnT1) that cadmium exposure leads to a change in the steady state protein expression parallel to that of mRNA. The authors should also provide immunoblot data for ZnT1 for the zinc, BSO and Aza-dC treatment experiments.

Response: We thank the Reviewer for her/his comment and suggestion concerning assessment of protein expression levels especially for zinc transporters that may potentially modulate Cd toxicity such as ZnT1, ZIP8 and ZIP14.  The suggested lines of investigation are in progress in our laboratory and results will be reported in a future paper.  At least, Western blot analysis showing expression of ZIP8 at protein level has been reported (ref. 31). Our paper reports quantitative gene expression (copies per 1000 β-actin transcript) for nearly every zinc transporter discovered in a cell culture model of urothelium. It is first report of possible epigenetic regulation of zinc transporter other than ZIP8 that has been shown by other research group. To better reflect the value of our quantitative gene expression data presented in this report, we have inserted statements in the Discussion (lines 164-169). The Discussion section has undergone extensive revisions and we have added three more references (ref. 31-33). Please see response to point 2 below.

Point 2: It appears from the manuscript that the authors are suggesting that cadmium toxicity (by itself, or in presence of zinc/BSO/Aza-dC) is modulated by changes in zinc transporter expression and consequently on the concentration of intracellular cadmium. The authors need to measure the level of intracellular cadmium under each of these conditions to empirically validate this hypothesis.

Response: We agree with the Reviewer that quantitation of intracellular Cd concentration, which is an integrative function of efflux and influx zinc transporters, would validate our hypothesis. However, we know of at least four reports in which Cd accumulation levels were measured and were related to resistance/sensitivity to Cd toxicity (ref. 21, 22, 32 and 33).  In revision we have discussed these previous works in details (lines 273-285).  Most importantly, epigenetic silencing of the ZIP8 gene by DNA hypermethylation has been shown to be a plausible molecular mechanism by which the ZIP8 gene is down-regulated leading to a reduction in Cd accumulation and tolerance (ref. 33).

Round 2

Reviewer 2 Report

  1. Regarding my previous comment #1, please clearly state the significance of focusing on a urinary bladder epithelium model in this research.
  2. Regarding my previous comments #3 and 5, if the batch-to-batch variation is as large as shown in the manuscript, then make sure the following points and clearly describe them in the manuscript: 2-1) Indicate how many biological and technical repeats were performed for each experiment and figure. 2-2) Each graph in Figures were prepared by the data collected from the identical batches.
  3. Regarding my previous comment #4, please indicate unassessed data points in the Figure.

Author Response

Reviewer 2

We thank the Reviewer for the guidance given to improve our paper. We provide below point-by-point response in accordance with the additional suggestions. We hope we have fully addressed concerns and issues raised.

Point 1: Regarding my previous comment #1, please clearly state the significance of focusing on a urinary bladder epithelium model in this research.

Response: We have inserted statements (lines 64-68) and one additional reference has been included (ref. 22).  We have rephrased the last sentence of Introduction to reflect how our study differed from others (lines 73-75).  We wish to reiterate that we are first to have examined multiple zinc transporters (influx and efflux transporters included) as the molecular basis of Cd tolerance, while other previous works examined a single transporter such as ZIP8 or ZnT1 only. The intracellular concentration of Cd is an integrative function of influx/efflux transporters.

Point 2: Regarding my previous comments #3 and 5, if the batch-to-batch variation is as large as shown in the manuscript, then make sure the following points and clearly describe them in the manuscript.

Point 2.1: Indicate how many biological and technical repeats were performed for each experiment and figure. 

Response: In the original manuscript, we have indicated the number of cell cultures used for each Cd concentration in Table 1 footnote. In a revised version of manuscript, the statement has been moved to Materials and Method Section to explain number of repeats (lines 88-92). Another reference has been included to indicate that expression of ZIP and ZnT genes by UROtsa cells at basal state were assessed using the Affymetrix 133 Plus 2 microarray.

Point 2.2: Each graph in Figures were prepared by the data collected from the identical batches.

Response: We have indicated identical or non-identical cell batches in the legends of Fig. 1 (lines 177-178), Fig. 2 (lines 195-196, Fig 3 (lines 221-223) and Fig. 4 (lines 253-254).

Point 3: Regarding my previous comment #4, please indicate unassessed data points in the Figure.

Response: As advised, we have now inserted, “An effect of 48-h exposure was not assessed for 0.1 and 1.5 µM Cd concentrations." in Figure legend of Figure 1 (lines 177-178).

Reviewer 3 Report

The authors have satisfactorily responded to the issues raised previously. The manuscript is acceptable for publication.

Author Response

Reviewer 3

The authors have satisfactorily responded to the issues raised previously. The manuscript is acceptable for publication.

Response: We thank the reviewer for the guidance given and for accepting our paper.  

Round 3

Reviewer 2 Report

Many of my comments have been addressed. The manuscript should be ready for publication.